# Research on Green Space Service Space Based on Crowd Aggregation and Activity Characteristics under Big Data—Take Tacheng City as an Example

**DOI:** 10.3390/ijerph192215122

**Published:** 2022-11-16

**Authors:** Tai Zhang, Bin Wang, Yisong Ge, Chengzhi Li

**Affiliations:** 1College of Ecology and Environment, Xinjiang University, Urumqi 830046, China; 2Key Laboratory of Oasis Ecology of Ministry of Education, Xinjiang University, Urumqi 830046, China; 3Xinjiang Jinghe Observation and Research Station of Temperate Desert Ecosystem, Ministry of Education, Jinghe 833300, China

**Keywords:** crowd activities, crowd aggregation, point of interesting, green space accessibility, greenland services

## Abstract

People-oriented planning has become the mainstream of urban space design. As an important research object of urban space, the accessibility and service level of accessibility and service level of green space as important indicators to evaluate the level of urban livability cannot be truly fed back to people’s daily life. Therefore, based on big data and from the perspective of crowd activities and aggregation characteristics, this study analyzes the shortage of green space service space in Tacheng City and puts forward suggestions for improvement. The main conclusions are as follows: (1) The satisfaction of green space based on service scope covers up the imbalance of green space resources enjoyed by actual crowd activities and aggregation. (2) Although the accessibility of green space obtained by population density meets the needs in space, it cannot take care of the potential needs generated by daily crowd activities and aggregation, which leads to the overall spatial imbalance of accessibility. (3) The comprehensive analysis shows that the northeast and southwest regions are the focus of the later planning and construction. The southwest region echoes with the old urban area and attracts people’s daily activities. The woodland in the northeast region, as the main green space supply, meets the potential needs of the daily population activities and aggregation of the new development urban area and the old urban area, and also serves as a place for rest and entertainment to meet the needs of the activities and aggregation of the accidental behavior of the people in the new and old urban areas after the opening up.

## 1. Introduction

Chinese cities are undergoing profound development and transformation. The transformation of urban function, the continuous growth of the population, the rapid suburbanization of urban living space and the quality of life of residents are closely related to urban green space, and the service level of urban green space is not only affected by its spatial configuration, but also closely related to the daily activities and aggregation of urban residents.

Initial scholars mainly used the statistical index method, travel distance or travel cost method (simple buffer method/cost-weighted distance method/network analysis method), minimum distance method and gravity model method to analyze urban green space [1]. Then, scholars introduced gradient analysis [2], space syntax [3], and the Gauss two-step mobile search method [4], and through these methods roughly calculated the accessibility of green space. Then, models such as the decision support model, Huff model, proximity model, container model, gravity model and two-step floating catchment area model, etc., were used [5,6,7,8,9]. Additionally, through continuous simulation verification to improve its accuracy, because the traditional method less considers the actual accessibility and fairness of green space, on this basis, scholars introduced elements such as the following: fairness concept [10,11,12], social equity perspective [13], environmental justice perspective [14], supply and demand balance perspective [15,16], supply and demand matching perspective [17] and aging society perspective [18] to analyze the balance between population demand and green space service supply. With the development of information and communication technology and Internet of Things technology, the advancement of data disclosure, and the breakthrough and application of data acquisition, processing and visualization technology, the world has entered the era of networked big data [19]. Big data have higher accuracy, wider coverage and stronger timeliness than traditional data. In recent years, scholars have used spatial location data (Baidu thermal map [20], point of interest (POI) data [20,21,22], social network data [23], public participation platform data [24]) and behavior trajectory data (mobile phone positioning data [25,26], traffic sensing data [27] and sports social data [28]) to analyze the utilization of green space, green space services and public participation. These scholars have conducted various studies on the evaluation system and spatial distribution of urban green space from different perspectives and data sources. Although there have been some explorations in the analysis of green space by using big data in recent years, the research on green space service space by using big data is still in its infancy, and the research on urban green space service in different construction areas is also just beginning. Big data only consider the coverage level of space and do not emphasize the relevance of green space based on people. The interaction structure between urban green space and people is only quantified [29]. Additionally, peoples’ lives in the city will inevitably show the activity and aggregation characteristics, and the study of population activity characteristics is helpful to provide reference in green space planning and design; on the other hand, the study of population agglomeration characteristics helps to optimize the spatial layout and leave room for development. Therefore, this paper, from the perspective of population aggregation and activity characteristics of urban green space comprehensive service space to make up for the lack of a people-oriented concept of green space resource planning analysis, but is also more suitable for peoples’ daily lives on the fairness of green space research.

In this paper, taking the built-up area of Tacheng City as the research area, the concept of a “15-minute community life circle” is introduced, and the daily activities and aggregation characteristics of the population are identified by big-data POI data. The population density data are used to characterize the aggregation characteristics of the basic population, and the green space accessibility is calculated by the Gauss two-step moving method to characterize the population demand. Three different urban green space service radius calculation methods are selected to construct green space service space to characterize the supply of green space. After processing, the superposition analysis is carried out. This paper analyzes the distribution characteristics of green space and the enjoyment of green space services under human aggregation and activities, judges its distribution deficiencies and puts forward suggestions.

## 2. Materials and Methods

### 2.1. Study Area

Tacheng City is located on the northwest border of China and the northwest of the Xinjiang Uygur Autonomous Region, which is located at 82°41′–83°41′ E and 46°21′– 47°14′ N. It belongs to the temperate continental arid region. The scope of this study is the built-up area of Tacheng City (Figure 1).

### 2.2. Data Source and Processing

#### 2.2.1. Urban Green Space Data Sources

Based on the green space data in the planning scope of “Tacheng City Green Space Planning” and combined with field investigation, the green space distribution map was made with satellite images as the basic data of urban green space (Figure 2). According to the “classification standard of urban green space (CJJ/T85-2017)” (https://www.mohurd.gov.cn/ accessed on 1 April 2022), Tacheng green space is divided into park green space and protective green space. The service functions of different types of urban green space are shown in Table 1.

#### 2.2.2. POI Data

The POI data contain the location and attribute information of various hot spots related to human life, and their plane aggregation degree is proportional to the heat of life in the region, which can be used for urban functional zoning and identification [30,31,32]. In this study, the POI data source is the Bige Map GIS Office downloader Gaode electronic navigation map data(http://www.bigemap.com/, accessed on 1 April 2022), using interest point layer data. According to the characteristics of urban residents’ daily life, 19 types of data such as life service, accommodation service and public facilities are selected as samples to retain their attribute information such as latitude, longitude and name type. After statistical de-reprocessing, the number of interest points in the study area is 1294, which can represent the daily activities and aggregation characteristics of residents in the urban area.

#### 2.2.3. Population Data

The population size in the region directly raises the demand for green space, and the fineness of residents’ enjoyment of green space mainly depends on the resolution of population data [33]. At present, there are a variety of public large-scale population grid data, such as the GPW dataset produced by the area weighting method [32], China’ s population spatial distribution kilometer grid dataset produced by the multi-source geographic information fusion method and the WorldPop world population grid dataset (WorldPop data). WorldPop data are often used as a control group to verify product accuracy, which has high applicability and high spatial resolution, as well as population fitting accuracy in China [34,35,36]. The population data used in this study are downloaded from WorldPop’ 2020 WorldPop population dataset with a resolution of 100 m, and used as the basic population data for accessibility analysis and the characterization of population-based aggregation characteristics.

#### 2.2.4. All Input Data Summary

To visualize the type and source of input data in this study, they are summarized in Table 2.

### 2.3. Methods

#### 2.3.1. Residents’ Living Heat Level

The life heat of residents in the study area was divided by nuclear density estimation. Using the ArcGIS10.4 Spatial Analyst Tools, the calculated point density plane can characterize the distribution characteristics of residential heat. The position of the density to be calculated is set to s, and the nuclear density at s is calculated by the following formula:(1)$$f\left(s\right)=\overset{n}{\underset{C_{i}\leq 1}{\sum }}\frac{1}{\pi h^{2}}\left(1-\frac{{\left(s-c_{i}\right)}^{2}}{h^{2}}\right)$$
where h denotes the search radius; n denotes the distance from the position; and $s-c_{i}$ denotes the distance between the density position s to be calculated and the core element $c_{i}$.

#### 2.3.2. Green Space of Service

##### (1) Cooling Effect

The photosynthesis and transpiration of vegetation in urban green space can reduce air temperature and improve regional microclimate. There is a logarithmic relationship between green space area and service radius. There is a logarithmic relationship between urban green space area and the cooling effect, and there is also a logarithmic relationship between green space area and the service radius. According to the research of Su Yongxian et al. [37]., the formula of the cooling service radius (buffering radius) through green space area is as follows:(2)y=53.668lnx−448.33
where y denotes the maximum cooling service radius (buffering radius); x denotes the green area.

Using the “Buffer” tool in ArcGIS 10.4, the service radius of mitigating the heat island effect is established according to the calculation results of the formula, and the distribution of the functional service space in the study area is obtained.

##### (2) Rest and Recreation

Urban park green space can provide a safe and quiet recreational environment for residents, and guide people to have a comfortable and positive mood, which is beneficial to the physiological and mental health of users [38,39]. Since the study area is only 45.8 km^2^, this study refers to the design of the community life circle, considers the cost of residents’ walking to urban green space, uses the buffer method, takes the pedestrian speed of 5 km/h as the calculation standard, and takes each urban park as the center to establish buffers at two scales of distance h ≤ 420m (walking time < 5 min) and 420 m < h ≤ 1250 m (walking time is about 5–15 min), and then uses the Cost Distance tool in ArcGIS to set the radius of the urban park green space recreation service.

##### (3) Disaster Prevention

Urban green space has the service function of disaster prevention and shelter, which can protect public life safety and health [40]. The radius of disaster prevention and avoidance service in urban green space varies with its scale and type. Refer to Wu Jiansheng and Li Shaoling et al. to determine the radius of urban green space (buffering radius) disaster prevention and avoidance services [41,42] (see Table 3).

Using the Buffer Wizard tool in ArcGIS, according to the buffer distance of the disaster prevention and avoidance service radius of similar urban green space, the overlap part of the buffer is fused to obtain its service space.

#### 2.3.3. Green Space Accessibility

This paper selects the accessibility evaluation based on the Gaussian two-step mobile search method and introduces the concept of the 15-minute community life circle. The search threshold is set to 1250 m (walking for 15 min). The two-step mobile search method considers both supply and demand factors, which can comprehensively and easily calculate the accessibility of park green space. Among them, using the Gaussian function to establish spatial attenuation rules is the most commonly used method in various expansion forms [43,44]. In this paper, the improved Gaussian two-step moving search method is used for accessibility evaluation. The specific steps are as follows:(3)$$G\left(d_{ij}\right)=\frac{e^{-\frac{1}{2}\times \left[\frac{d_{ij}}{d_{0}}\right]}-e^{-\frac{1}{2}}}{1-e^{-\frac{1}{2}}}$$
where G(d_ij_) denotes the Gaussian attenuation function considering the spatial friction problem; i denotes the demand point; j denotes the geometric center point of park green space; d_ij_ denotes the straight-line distance from the demand point to the geometric center point of park green space;d_0_ denotes the limit distance of the road network for people to park in green space.
(4)$$R_{j}=\frac{S_{j}}{\sum_{k\in \left\{d_{kj}\leq d_{0}\right\}}G\left(d_{ij}\right)D_{k}}$$
where R_j_ denotes the supply–demand ratio; D_k_ denotes the population of each demand unit k; d_kj_ denotes the road network distance between positions k and j; S_j_ denotes the area of park green space j.
(5)$$A_{I}{}^{D}=\underset{j\in \left\{d_{i}\leq d_{0}\right\}}{\sum }G\left(d_{ij}\right)R_{j}$$
where A^I^_D_ denotes the accessibility of park green space based on time cost.

## 3. Results

### 3.1. Distribution Characteristics of Life Heat in Population

The interest points in the study area are concentrated in the central region of the old urban area, showing an irregularly decreasing density distribution from the central urban area of the old urban area to the outer urban area (Figure 3). It is divided into five density zones: Cold area, Sub-Cold area, Not Significant area, Sub-Hot area and Hot area. The occupied areas were 1.05 km^2^, 1.78 km^2^, 3.32 km^2^, 6.59 km^2^ and 35.60 km^2^, accounting for 2.17%, 3.68%, 6.87%, 13.64% and 73.64%, respectively.

From the perspective of the distribution characteristics of people’s heat of life, the heat of production and life outside the old city is low, forming multiple small-scale agglomerations. In terms of distribution, the heat of production and living in the northern city is higher than that in the southern city, and the heat in the western city is slightly higher than that in the eastern city. In addition to the intensive distribution of production and living heat in the old urban area, the central circle-type heat distribution has not been formed.

### 3.2. Spatial Distribution Characteristics of Urban Green Space Cooling Effect Service

The service space of the cooling effect of green space in the study area is 25.237 km^2^, accounting for 55.09% of the study area; the service blind area is 20.537 km^2^, accounting for 44.91% of the study area. At the same time, there are also situations in the study area where the green space is too close and the service space overlaps. The overlap area is 2.331 km^2^, accounting for 9.24% of the green space service area. The best cooling effect area of urban green space is located in the Sub-Cold area, and its actual service space is 3.939 km^2^, accounting for 63.86% of the density area. Although the actual service space of green space in the Not Significant area and Cold area is more than 50%, it is dominated by large-patch green space with low dispersion, which is not conducive to the heat exchange between urban construction land and green landscape. From the perspective of green space service supply, the spatial distribution of the mitigation heat island effect service in the study area is relatively unbalanced (Table 4).

### 3.3. Distribution Characteristics of Green Rest and Entertainment Services in Urban Parks

The radius distribution characteristics of recreation and entertainment service space in urban park green rest space are 420 m and 1250 m, and the service space can reach 12.09 km^2^ and 41.88 km^2^, accounting for 26.39% and 91.43% of the areas in the reclamation area, respectively. The service space radius of 1250 m has an overlapping area of 28.30 km^2^, accounting for 67.57% of the service space. The park service level in the Hot area is the best, and the actual service space of different service radii reaches 55.8 % and 99.98%, respectively, which basically meets the needs of residents in different living spaces in the area for recreation. In the 420 m service radius of urban green space, the actual service space covers only 43.72% of the Sub-Hot area, 49.30% of the Not Significant area, 30.74% of the Sub-Cold area and 18.97% of the Cold area. There are more than 50% of the service blind areas in these four areas. Therefore, the residents’ demand for recreational services within the built-up area does not match the spatial distribution of park green space (Table 5).

### 3.4. Spatial Distribution Characteristics of Disaster Prevention and Avoidance Service in Urban Green Space

The service range of disaster prevention and risk avoidance green space in the study area is large, and the total service space is 43.69 km^2^, accounting for 95.37 % of the study area. However, there is still an overlapping area of 39.71 km^2^ in the study area, accounting for 90.89% of the green space service area. In the Hot area, the actual service space of urban green space disaster prevention and avoidance accounts for 0.9936 km^2^, accounting for 99.98% of the area; in the Sub-Hot area and the Not Significant area, the actual service space of urban green space disaster prevention accounts for 100% of the two regions; in the Sub-Cold area, the actual service space for disaster prevention and avoidance in urban green space is 6.177 km^2^, accounting for 98.92% of the area; the actual service space of disaster prevention and risk avoidance in urban green space in the Cold area is 31.263 km^2^, accounting for 93.68% of the area. Since the urban area is relatively compact and there are many natural green rivers, the disaster prevention and risk avoidance service ability is strong, so that the coverage area of the disaster prevention and risk avoidance green space service is more than 90%. The spatial distribution of the disaster prevention and risk avoidance service near the large park green space in the study area is reasonable, but the green space in the high-density area is small and the distribution is not concentrated, which fails to meet the demand for disaster prevention with dense population activities (Table 6).

### 3.5. Spatial Distribution Characteristics of Urban Green Space Integrated Function Service

The service space and life heat map of three service functions of urban green space in the study area, including heat island effect mitigation, recreation and entertainment, disaster prevention and risk avoidance, were superimposed and analyzed. The statistical results are visible (Table 7).

It can be seen from the table that the comprehensive service space of green space in the study area is mainly distributed in the northeast, followed by the southeast and northwest, and the southwest is the worst, with a total area of 24.89km^2^, accounting for 54.33%. In the five types of living hot areas, the coverage of comprehensive green space service space in the Hot area is the largest, accounting for 62.84% of the density area, while in the Cold area, only 47.84% of the areas enjoy three kinds of green space services, and there are more than 30% blind areas of comprehensive service in the five areas.

### 3.6. Accessibility Characteristics of Green Space

The higher the accessibility, the easier it is for the region to enjoy park green space resources. From the contribution of park green space accessibility value, the high accessibility of the overall level of the study area is mainly derived from the large number, large area and wide distribution of natural forest green space. There are 233 units in the study area, of which 21 reachability values are greater than the per capita park green space, accounting for 10 of the total number of units, mainly distributed in the eastern part of Tacheng City. There are 126 units with medium, mid-high and high accessibility, accounting for 54% of the total number of units, which are mainly distributed in the central, western and northeastern regions. There are more forest green spaces in this region, and the coverage area is large. Several strip parks pass through the city, with more adjacent communities and developed road networks, and their green space accessibility is better (Figure 4).

### 3.7. Comprehensive Analysis

According to the distribution of residents’ heat of production and living, the comprehensive function of green space service space, accessibility, supply and demand characteristics and OD cost and other factors, the people’s heat of production and living, the comprehensive service space of green space and accessibility are overlapped. After processing, eighteen partitions can be obtained, which are divided into life density (low, medium and high), accessibility (low, medium and high) and green space service (whether to enjoy). In this paper, 18 districts are integrated with green space supply capacity, life heat and whether to enjoy services. Finally, six districts are merged, as shown in Table 8.

According to the characteristics of residents’ life, the six areas are named as: High-quality living area, Livable area, Underdeveloped area, Green space service blind area, Under supply and demand area and Comprehensive blind area. The comprehensive analysis diagram can be seen in Figure 5.

## 4. Discussion

In the high-quality living area with high population density and more daily activities of the population, there is a blind area with low accessibility and uncoverable green space. This area is located in the core area of the development of the old urban area, which may be due to the poor awareness of human ecological environment and low attention to green space in the early stage of urban construction in study area, leading to the phenomenon that the rapid development of the city ignores the construction of green space and destroys green space to construct buildings. As the center of the development of the old city, there are more jobs in this area, so it plays the role of the employment center of the surrounding villages and towns, resulting in a high density of population activities, coupled with its living facilities such as large shopping centers and food streets and other crowd gathering points; the attraction of population activities leads to a very high potential demand for green space in this area, but due to the shortage of land in the central city, there are few land areas available for transformation or expansion of green space, resulting in the existing central city service coverage, insufficient supply and unbalanced green space resources. At present, the normalization of epidemic prevention and control can improve the ability of cities to respond to major emergencies by improving community resilience. The lack of green space in the central urban area highlights several points, which can reduce the scale of green space in the central urban area of land resources shortage to the level of community parks. Green space plays a vital role as a community isolation and buffer space and an activity gathering point for residents to improve personal resilience and community cohesion. The supply mode of new community parks such as open community green space and aerial gardens can be used as a supplement to green space.

The livable areas and the three blind areas show a trend of gradual change from northeast to southwest as a whole, which coincides with the characteristics of population activities. At the same time, it also shows that the service and supply of urban green space are not balanced at the level. The reason may be that the natural forest land along the river in the northeast of the original landform urban area of Tacheng spreads from northeast to southwest in a band, while there are more barren areas in the southwest, which affects the activities of the initial population. With the urban construction and the change of people’s ideas on the ecological environment, the southwest is the focus of urban expansion and construction in the later period. It is not only the core area connecting with the northeast woodland river to become a green corridor, but also the ecological civilization construction area that shares urban pressure and supplements the green space supply of dense urban agglomerations. It echoes with the old urban areas, improves the green space supply and service, and promotes the daily population flow and communication in the new and old urban areas. However, in the northeast natural forest area, the daily population activities are less, the population density is low, and most of them are villages where the elderly live. However, due to the large area and sufficient supply of natural river forest land, it will make the residents of the new and old cities in southwest China have the activity characteristics under accidental behavior, such as holiday visiting relatives and friends or picnic parties.

There are some shortcomings in this study. Firstly, the connotation of urban community life circle is rich, and the mode of human activities is complex. This paper only refers to the elements of a community life circle as a reference and refers to the range of community life circles as a search threshold to calculate, which cannot fully represent the crowd activities in various activity spaces. Secondly, although the data used in this paper are relatively new and have certain timeliness, only the characterization of supply and demand of park green space and residential population is considered in the modeling analysis, and the difference in service supply capacity under the influence of park internal facilities and ecological environment is not considered. The reachability only takes the centroid of the TAZ unit, without further subdivision. Using population density as a basis for statistics, short-term resident population and actual service population data are not calculated and urban population movements are ignored; the lack of real travel modes (bus, self-driving, bicycle, walking, etc.) and route data makes the activity characteristics different from the actual situation. Finally, small-scale urban green space is an important carrier to highlight regional characteristics, and also a place to meet the spiritual and cultural needs of residents. Future research can further analyze the above factors combined with the perspective of environmental justice to think and analyze, and to put forward planning suggestions to better serve the “people-oriented” smart city.

## 5. Conclusions

Based on green space accessibility and green space service, this study uses the WorldPop population dataset to characterize the basic aggregation characteristics of the population and uses the POI data to characterize the daily activities and aggregation characteristics of the population. It analyzes the interaction between population activities and aggregation characteristics and green space service space, identifies the imbalance between green space resources and daily supply and demand of the population in urban planning, and puts forward suggestions for improvement of its planning and design deficiencies. It also provides ideas for the application of big data in other urban planning fields.

The main conclusions are as follows:

The coverage of green space services and daily activities and aggregation characteristics of the population showed a trend of Sub-Hot area > Hot area > Sub-Cold area > Not Significant area > Cold area. The actual service space of urban green space in crowd activity and aggregation low density area is large, but the coverage rate is the lowest. The satisfaction of green space based on service scope covers up the imbalance of green space resources enjoyed by actual population activities and aggregation. With the increase of population activities and aggregation in low-density areas in urban construction, the problem will become increasingly prominent.The overall level of green space accessibility based on population-based aggregation is good, but there is an obvious irregular increasing trend from the old urban area to the outside, which cannot meet the potential needs of people’s daily activities and aggregation, resulting in the imbalance of accessibility space;Through comprehensive analysis, it can be seen that the northeast and southwest regions are the focus of later planning and construction, and the southwest region and the old urban area echo each other and attract people’s daily activities. The forest land in the northeast region, as the main green space supply, meets the potential needs generated by the daily population activities and aggregation of the new development urban area and the old urban area, and also serves as a place for rest and entertainment to meet the needs of the activities and aggregation of the people with occasional behaviors in the new and old urban areas after opening up.

## Figures and Tables

**Figure 1 ijerph-19-15122-f001:**
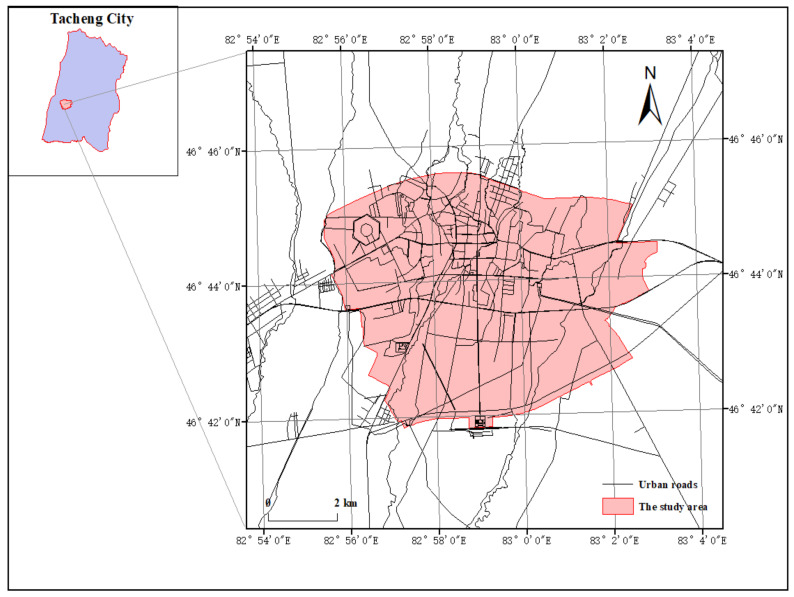
Geographical scope of the research.

**Figure 2 ijerph-19-15122-f002:**
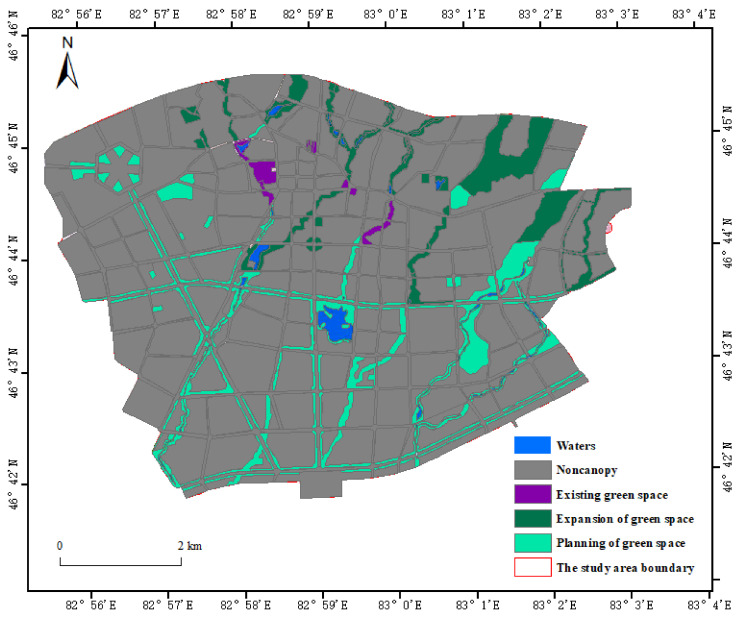
Basic data map of green space in the study area.

**Figure 3 ijerph-19-15122-f003:**
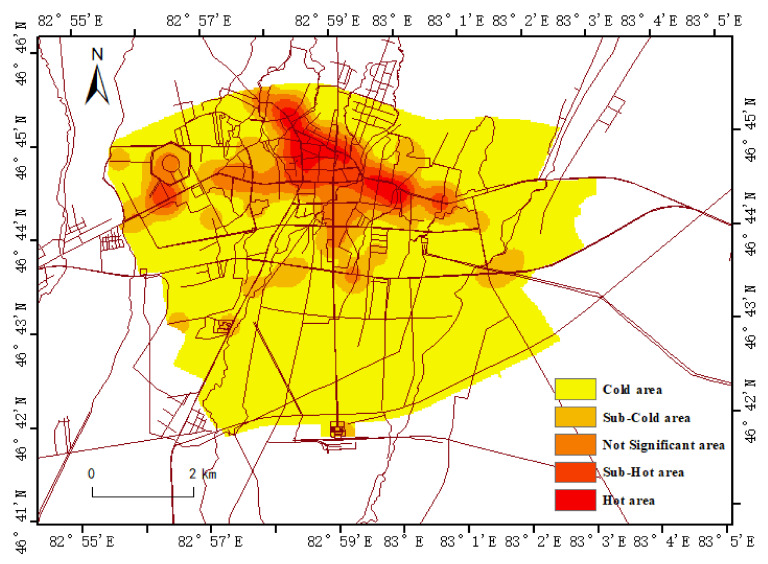
Life characteristics map of the population in the study area.

**Figure 4 ijerph-19-15122-f004:**
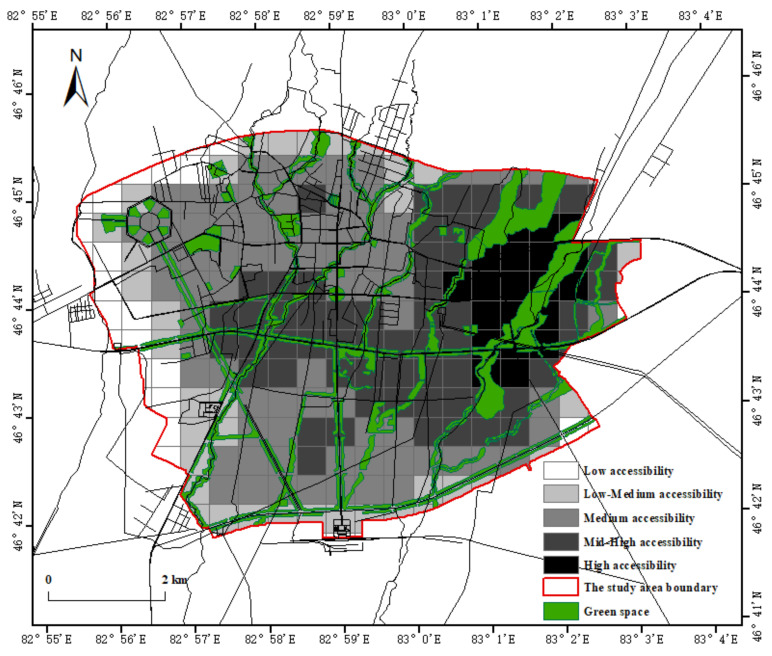
Green space accessibility analysis chart.

**Figure 5 ijerph-19-15122-f005:**
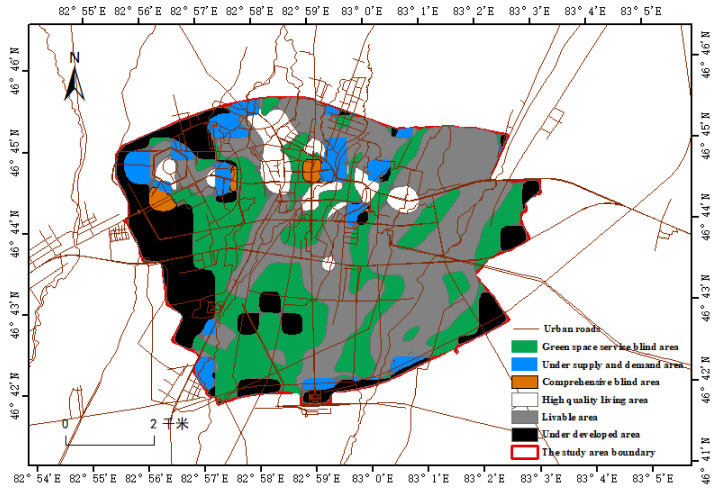
Comprehensive analysis chart.

**Table 1 ijerph-19-15122-t001:** Services provided by different types of urban green space.

Green Space Types	Cooling Effect	Rest and Recreation	Disaster Prevention
Park green space	√	√	√
Green buffer	√ ^1^	— ^2^	√

^1,2^ √ indicates that green space has this function, — indicates that green space does not have this function.

**Table 2 ijerph-19-15122-t002:** Input data source summary table.

Input Data	Data Type	Data Sources
Study area range	vector data	Geospatial Data Cloud (http://www.gscloud.cn/, accessed on 1 April 2022)
Green space data	vector data	The vector map drawn by combining remote sensing imaging and Tacheng urban green space system planning
POI data	vector data	Bige Map GIS Office (http://www.bigemap.com/, accessed on 1 April 2022)
Population data	raster data	WorldPop official (websitehttps://www.worldpop.org/, accessed on 1 April 2022)

**Table 3 ijerph-19-15122-t003:** Greenland disaster prevention service radius table.

Green Space Types	Scale	Service (Buffering) Radius
Park green space	0.5–1 hm^2^	0.5 km
	1–10 hm^2^	1 km
	10–50 hm^2^	2 km
	>50 hm^2^	3 km
Green buffer	>0.5 hm^2^	0.3 km

**Table 4 ijerph-19-15122-t004:** Service space statistics of green space alleviating the heat island effect in each living heat area.

Type	Cold Area	Sub-Cold Area	Not Significant Area	Sub-Hot Area	Hot Area
Area/km^2^	33.735	6.246	3.148	1.687	0.994
Service area/km^2^	17.664	3.989	1.889	1.072	0.625
Ratio/%	52.36	63.86	60.02	63.54	62.82

**Table 5 ijerph-19-15122-t005:** Service space statistics of green space alleviating the heat island effect in each living heat area.

Type	420 m Service Radius		1250 m Service Radius	
	Service Area/km^2^	Ratio/%	Service Area/km^2^	Ratio/%
**Cold Area**	6.7539	18.97	29.847	83.85
**Sub-Cold Area**	2.0262	30.74	5.869	89.05
**Not Significant Area**	1.6428	49.30	3.262	97.89
**Sub-Hot Area**	0.7782	43.72	1.780	100
**Hot Area**	0.5855	55.82	1.0488	99.98
**Total**	12.09	26.39	41.883	91.43

**Table 6 ijerph-19-15122-t006:** Green Space for Disaster Prevention and Avoidance Services in Hot Living Areas.

Type	Cold Area	Sub-Cold Area	Not Significant Area	Sub-Hot Area	Hot Area	Total
**Area/km^2^**	33.724	6.244	3.147	1.686	0.9938	45.81
**Service Area/km^2^**	31.263	6.177	3.147	1.686	0.9936	43.69
**Ratio/%**	93.68	98.92	100	100	99.98	95.37

**Table 7 ijerph-19-15122-t007:** Statistics of Comprehensive Service Space of Green Space in Hot Living Areas.

Type	Cold Area	Sub-Cold Area	Not Significant Area	Sub-Hot Area	Hot Area	Total
**Area/km^2^**	0.994	1.686	3.148	6.244	33.724	45.81
**Service Area/km^2^**	0.625	1.056	1.889	3.864	16.134	24.89
**Ratio/%**	62.84	63.53	60.01	61.89	47.84	54.33

**Table 8 ijerph-19-15122-t008:** Integrated partition table.

Serial Number	Green Space Supply Capacity	Life Heat	Green Space Service
1	Supply ≥ Demand	Not significant, Sub-hot and Hot	Enjoy
2	Supply ≥ Demand	Sub-cold and Cold	Enjoy
3	Supply ≤ Demand	Sub-cold and Cold	Not enjoying
4	Supply ≥ Demand	Not significant, Sub-hot and Hot	Not enjoying
5	Supply ≤ Demand	Not significant, Sub-hot and Hot	Enjoy
6	Supply ≤ Demand	Not significant, Sub-hot and Hot	Not enjoying

## Data Availability

Data are contained within the article.

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
