# Peer review of "Research on Green Space Service Space Based on Crowd Aggregation and Activity Characteristics under Big Data—Take Tacheng City as an Example"

_ijerph, 2022, doi:10.3390/ijerph192215122_

Round 1
Reviewer 1 Report
Metodology of the research must be detailed described. Describe geostatistical methods which will be used to provide results described in Results section.
Also describe input parameters. Do you use raster or vector data or both in your analysis. Put table with all input data used in this research.
Describe how did you get formula 2? What are the parameters of the buffer. Which radius are you buffering and why. It is unclear now.
Reviewer 2 Report
The original research article on "Research on Green Space Service Space Based on Crowd Aggregation and Activity Characteristics Under Big Data" by Yisong Ge et colab is very well written and presented.
The introduction is descriptive enough and the results are well structured. I have no comments for further improvement and recommend the publication in present form.
Reviewer 3 Report
Review on the original article "Research on Green Space Service Space Based on Crowd Aggregation and Activity Characteristics Under Big Data" by authors Tai Zhang, Chengzhi Li, Bin Wang, Yisong Ge.
I found that the authors have done a very good job, as the paper is well structured.
Apart from one very minor remark regarding the abbreviations, I recommend the publication of this paper.
Please take care about abbreviations (first define full name and then abbreviation). Check in whole text.
Congratulation on your work!
